# An Integrated Approach to Assess the Environmental Impacts of Large-Scale Gold Mining: The Nzema-Gold Mines in the Ellembelle District of Ghana as a Case Study

**DOI:** 10.3390/ijerph18137044

**Published:** 2021-07-01

**Authors:** Dawuda Usman Kaku, Yonghong Cao, Yousef Ahmed Al-Masnay, Jean Claude Nizeyimana

**Affiliations:** 1School of Environment, Northeast Normal University, Changchun 130117, China; dawd642@nenu.edu.cn (D.U.K.); yues271@nenu.edu.cn (Y.A.A.-M.); nizeyeclaude570@gmail.com (J.C.N.); 2State Environmental Protection Key Laboratory of Wetland Ecology and Vegetation Restoration, School of Environment, Northeast Normal University, Changchun 130117, China

**Keywords:** Ghana, Ellembelle district, large-scale gold mining, environmental impacts, normalized difference vegetation index (NDVI)

## Abstract

The mining industry is a significant asset to the development of countries. Ghana, Africa’s second-largest gold producer, has benefited from gold mining as the sector generates about 90% of the country’s total exports. Just like all industries, mining is associated with benefits and risks to indigenes and the host environment. Small-scale miners are mostly accused in Ghana of being environmentally disruptive, due to their modes of operations. As a result, this paper seeks to assess the environmental impacts of large-scale gold mining with the Nzema Mines in Ellembelle as a case study. The study employs a double-phase mixed-method approach—a case study approach, consisting of site visitation, key informant interviews, questionnaires, and literature reviews, and the Normalized Difference Vegetation Index (NDVI) analysis method. The NDVI analysis shows that agricultural land reduced by −0.98%, while the bare area increases by 5.21% between the 2008 and 2015 periods. Our results show that forest reserves and bare area were reduced by −4.99% and −29%, respectively, while residential areas increased by 28.17% between 2015 and 2020. Vegetation, land, air, and water quality are highly threatened by large-scale mining in the area. Weak enforcement of mining policies, ineffective stakeholder institution collaborations, and limited community participation in decision-making processes were also noticed during the study. The authors conclude by giving recommendations to help enhance sustainable mining and ensure environmental sustainability in the district and beyond.

## 1. Introduction

Mineral wealth is an essential asset to enhance economic growth and foster infrastructure development, including building schools, hospitals, and road networks. The mining sector has gained wide recognition for being a blessing to most African and other developing nations with abundant mineral resources. Contrary, the sector is noticed to cause environmental issues, such as water and air pollution, land degradation, destruction of vegetation, and harm to biodiversity and other habitats [1,2,3].

In Ghana, mining is the fourth largest economic activity, and it accounts for about 5.5% of Ghana’s gross domestic product (GDP) and employs almost 300,000 people [2]. Ghana’s mineral production increased between the 2015 to 2018 production periods. An increase in the country’s gold production (4%) benefited from several factors, including the scheduled output increase at two Newmont Gold Corp Mines at Ahanfo and Akyem [4,5]. The Ghana Revenue Authority (GRA) in 2016 reported that the mining sector regained its position as the dominator of a source of direct domestic revenue in 2015. Total revenue generated by the mining and quarrying sector increased from GHȻ 1.35 billion to GHȻ 1.65 billion in 2015 and 2016, respectively, standing for 22% growth [6].

Despite the significant contribution of the mining sector to the socio-economic development of Ghana, the related adverse impacts are on the rise and very alarming. The environment of most mining areas in the country is continuously deteriorating. Its immense economic value keeps diminishing from time to time, due to the heavy concentration of mining activities in such places [7,8]. Most mining activities possess detrimental effects on the host environment, due to their continued use of modern, sophisticated machinery, harmful chemicals, and extended blasting levels. Unsupervised and unsafe activities by large-scale and artisanal small-scale mines are characterized by overwhelming environmental degradation, including water, air, and noise pollution, deforestation, destruction and loss of vegetation and ecosystem, land degradation, loss of soil fertility, and massive erosion [8,9,10]. Agricultural lands are not only generally degraded, but very scarce. The scarcity status of arable land for agricultural production has also led to the shortening of the fallow period from 10–13 years to 2–4 years [7,11]. Large-scale gold mining activities have threatened the agriculture sector as the backbone of Ghana’s development.

However, in the wake of these increasingly environmental challenges posed by the mining sector, the small-scale mining sector receives much attention over large-scale mines. In Ghana, small-scale artisanal firms stood for frequent accusations as guilty of these environmental challenges, due to their unsafe modes of operations. This, therefore, led to the imposition of a ban on small-scale mining activities in Ghana by the government in 2017 [9,12,13]. Research works in this study domain primarily focused on small-scale mining activities and their impacts on the environment, mitigation measures, and regulations to sustain the small-scale mining sector. For example, Bansah et al. (2016), in their work, suggested that small-scale mining firms should utilize professional engineers in their operation to help control the related negative environmental and social impacts [1,12]. Boafo et al. (2019) also postulated that environmental consequences caused by gold mining in Ghana are mainly a result of small-scale mining firms and weak institutional supervision [9]. Besides, the environmental and social impacts of gold mining in Ghana within the last decade are very challenging, with less focus and minimal research on the large-scale gold mining sector.

As a result, this study aims to assess the environmental impacts of large-scale gold mining in Ghana with the Ellembelle district as a case study. Specifically, the goal of this study was to: (1) Assess the environmental impacts emanating from large-scale gold mining activities through field/site and community visitation; (2) use the Normalized Difference Vegetation Index (NDVI), and Land-use maps to analyze the vegetation and land-use changes in Ellembelle from 2008 to 2020, due to large-scale gold mining, and (3) recommend possible mitigation measures to enhance sustainable mining and ensures environmental sustainability in the district and beyond. 

### Overview of Gold Mining Activities in Ghana

Ghana’s rich gold mines and other mineral fields have attracted some leading mining companies in the world into the country, which has resulted in tremendous growth in the country’s mining sector for the past few years [14]. In Ghana, gold mining activities are in two main forms, which are small-scale and large-scale mining. Small-scale mining is well known in Ghana as “galamsey” (Figure 1A), which refers to “gather them and sell” [15]. This mining type requires less capital for its start-up and operations and is mainly undertaken by a small group of people with less technical skills [2,16]. Small-scale mining was restricted primarily to Ghanaians, but today, the activity is undertaken by both Ghanaians and foreigners, with an estimated number of 200,000 people engaged in [17].

Large-scale mining receives much attention from this paper. Large-scale mining is the opposite of small-scale mining, which embarks on mechanized and sophisticated equipment and employs highly trained, educated, and technical know-how personnel in mineral resources explorations [18,19]. It is the type of mining that is generally associated with multinational or multisite companies, embedded in global capital and finance markets, and part of the international supply of mineral and metals commodities [20]. This type of mining operation has a worldwide employment rate of about 2.5 million people and generates more than 95% of the world’s total mineral production [2]. Large-scale mining (Figure 1B) involves both deep pit and surface methods and requires huge capital investment. The use of vast portions of land, sophisticated machines, harmful chemicals, and the general extraction and processing methods, such as drilling, blasting, haulage of ore, crushing, and screaming agglomeration, makes it mandatory for large-scale miners to seek government approval and acquire a license before commencing operations [16,21]. Currently, in Ghana, about 23 large-scale mining companies and over 300 registered small-scale artisanal groups engage in mineral exploration [14,16]. Private individuals, mostly foreigners, own 90% of these companies and 10% free shares or 20% optional offered to the government [2]. Gold, diamond, bauxite, and manganese are the main minerals produced by large-scale mining companies in the country. Other industrial minerals, like silver, salt, and limestone, are produced by small-scale operators [14,22,23]. 

## 2. Materials and Methods

### 2.1. Study Area

The Ellembelle District is one of the 14 Metropolitan, Municipal, and District Assemblies (MMDAs) in the Western Region of Ghana, and it is located on the southern part of the region between longitudes 2°05′ W and 2°35′ W and latitude 4°40′ N and 5°20′ N [24]. It has a total area of 995.8 km^2^, representing about 9.8 percent of the landmass of the Western Region [25]. The district has an estimated population of about 86,501. It lies within the semi-equatorial climatic zone of the West African sub-region. The vegetation of the area is a moist semi-deciduous rainforest, but changes to secondary forest southwards as a result of human activities like logging and farming [25]. The underlying rock mainly consists Cambrian type of the Birimian formation and the Tarkwaian Sandstone-Association of Quartzite and Phylites types. This contains economic minerals, such as kaolin, silica, gold, and sandstone deposits [24]. The district is blessed with several rivers and streams; prominent among them is the Ankobra River, with its major tributaries like the Ahama and Nwini rivers [25]. The Ellembelle district is recognized as a resource-based district, due to the heavy extraction of gold and crude oil from the area. Nkroful, Teleku-Bokazo, Anwia, and Salman are dominants gold mining cities with small-scale gold mining activities occurring for almost three decades, while large-scale gold mining commenced in 2011 [24]. Mining has boosted economic development in the district, but has also caused severe damage to the regional eco-environment [25]. Figure 2 is a map of the Ellembelle District containing study communities.

#### 2.1.1. Case Study

##### Large-Scale Surface Mining in Ellembelle

Nzema Gold Mine is the operational name for the large-scale gold mining property in the Ellembelle district, and it is wholly owned by Adamus Resources Limited (Adamus), a Ghanaian subsidiary of Endeavour. Adamus holds four (4) mining licenses and eleven (11) prospecting licenses covering an estimated land area of 464 km^2^ that constitutes the Nzema property. Adamus has 90% interest in all its mining licenses, and 10% remains with the government of Ghana, as stated by the Mining Act of Ghana [26]. Salman ML, Telaku Bokazo ML, Nkrful ML, and Akango ML are the four mining leased sites operated by Adamus, and hold 100% interest in five granted prospecting licenses [26]. Figure 3 presents the Adamus Mineral License Perimeters, Site Layout, and Deposit Locations in the District. Today, large-scale mining is carried out as surface mining, where cyanidation is noticed to be the most prevailing technique employed by mining companies for non-sulphidic pale placer ore [23,27,28]. Large-scale mining in Ellembelle takes the form of conventional open pit mining methods, which encompasses drilling and blasting of competent material followed by loading and hauling. Drilling and blasting are carried out on 6 m to 15 m benches, depending on the proposed deposit’s geological scope and geotechnical settings as blasted materials are excavated in discrete 3 m high flitches [26].

### 2.2. Methods of Data Collection and Analysis 

The study is a case study approach which employed a double-phase mixed-method design to collect and analyzed data on the environmental impacts of large-scale gold mining in the Ellembelle District. Phase one consists of qualitative data acquisition through reviewing the relevant literature, field or site visitation, interviews, and questionnaires. Phase two involved remote sensing data that comprises the Normalized Difference Vegetation Index (NDVI) and Land-use Land-cover (LULC) map to critically analyze and detect the vegetation land-use changes in the study area caused by large-scale gold mining.

#### 2.2.1. Data Collection

##### Participant Observations, Interviews, and Questionnaires 

Field visitations and participant observations were done at some operational sites to allow researchers to be more abreast with mining activities that pose threats to the environment of mining host areas. Interviews and open questionnaires were administered to collect data from key informants from stakeholder institutions (District EPA officer, District Environmental health and safety officer, District planning officer, and mines workers) and opinion leaders from host communities with in-depth knowledge of the phenomena under study. Topics included in the interviews and questionnaire (Appendix A) were on the nature of the environment before the commencement of large-scale mining operations, impacts of mining activities on both the built and natural environment, environmental policies, and mitigation measures to curb environmental devastations exacerbated. Respondents were selected by using the purposive sampling methods. Secondary data was gathered from journals, articles, books, websites of the case company, and literature reviews.

##### The Normalized Difference Vegetation Index (NDVI) 

For the past decade, the NDVI differencing method and classification method are the most commonly used change detection methods that project detailed and accurate information for monitoring and determining LULC changes [29]. NDVI stands as one of the earliest remote sensing analytical products used to facilitate the complexities of multispectral imagery, is now the most popular index used for vegetation assessment. This popularity and widespread use relate to how an NDVI can be calculated with any multispectral sensor with a visible and a near-IR band. Kriegler et al. (1969) proposed a simple band transformation: Near-infrared (*NIR*) radiation minus red radiation divided by near-infrared radiation plus red radiation (Equation (1)), resulting in a newly simplified image called the Normalized Difference Vegetation Index (NDVI) [30]. The NDVI is widely used to detect the changes in vegetation at regional and global scales through reflective bands of satellite data. NDVI generated image is highly suitable to determine vegetation dynamics between two or more different dates [29,31,32]. NDVI image differencing is based on subtracting different time periods of the same location or scene. The NDVI image values can be simply deduced by subtracting the two or more different times pixel by pixel to create the differentiated image [33]. Lyon et al. (1998) investigated seven vegetation indices by using Landsat MSS image data for land cover change detection from three different dates and postulated that the NDVI differencing technique was the best vegetation change detection. Mathematically, the NDVI data layer is defined as [29,30,34]: (1)$$\mathrm{NDVI}\;=\frac{N_{IR}NIR-R_{ed}Red}{N_{IR}NIR+R_{ed}Red}$$
where NDVI is the normalized difference vegetation index. *Red* and *NIR* are spectral radiance (or reflectance) measurements recorded with sensors in red (visible) and *NIR* regions, respectively. Radiance (watts steradian^−1^ m^−2^ μm^−1^) is the measure of energy flux recorded by a sensor. The values of radiance are often rescaled to digital numbers (DN) as 6-bit or 7-bit (MSS), 8-bit (TM, ETM+), or 12-bit (Landsat8) unsigned integers. Reflectance is a unitless measure of the ratio of radiation reflected by an object relative to the radiation incident upon the object [35]. The NDVI values should be greater than zero (0) and must be between −1 and +1, where increasing positive values represent healthy and green vegetation, and negative values represent sparse or non-vegetated surface features like water, barren land, ice, snow, or clouds. The presence of atmospheric effects causes the (red and near-infrared) wavelengths to scatter, affecting the calculation of NDVI. The use of reflectance is important to control the effect of scattered radiation in the atmosphere. The changes in NDVI values among different objects are due to their relative changes in spectral responses. Therefore, Equation (1) can be changed, as follows [30,36]:(2)$$\mathrm{NDVI}=\frac{R-1}{R+1}$$
where NDVI is the normalized difference vegetation index. *R* is the ratio of *NIR* to Red, and is commonly referred to as the ratio vegetation index [30]. Equation (2) is more explicit for defining NDVI behavior patterns, due to different responses of *NIR* and Red to atmospheric effects [37].

To detect gold mining-related vegetation and LULC changes in Ellembelle, we collected three Landsat images of the study region (Landsat7, ETM+) image for 2008 and (Landsat8, OLI) images for 2015 and 2020 from archives of the United States Geological Survey (USGS). The ArcGIS software was used to process these images. At the first stage, the NDVI layers were generated, and band math was then performed on the resulting NDVI images by calculating the area of each map (2008–2015–2020). To determine the areas with land cover changes, the 2015 image values were deducted from the 2008 image values, and the 2020 image values were also deducted from the 2015 image values, respectively. To finally assess the changes in NDVI values, the NDVI-change image was sliced into the three classes, class 1, class 2, and class 3, showing areas with low, medium, and high NDVI density values, respectively. At the second stage, we investigated the changes in each vegetation and land cover type to find the areas where the NDVI values has changed and classified the resulting images using ArcGIS 10.3. An overlay comprising LULC maps of 2008, 2015, and 2020 was developed through the ArcGIS software, and a transition matrix was done for the overlaid LULC maps of 2008, 2015, and 2020.

#### 2.2.2. Data Analysis and Presentation

The content analytical method was used to analyze the qualitative data, while Excel (Microsoft, Redmond, DC, USA), ArcGIS (Esri, Redlands, CA, USA), and OriginLab software (OriginLab, Northampton, UK) analyzed available quantitative data. Outputs were presented as summary statistics in tables, graphs, maps, and pie charts. On-field pictures were also presented as figures to attest and support our results

## 3. Results and Discussion

### 3.1. Demographic Characteristics of Respondents

Table 1 shows the demographic characteristics of the respondents interviewed during the survey. A total of 65 people were sampled for the study, which were selected from the four mining host communities. Twenty respondents were selected from Nkroful as district capital, where respondents from all stakeholder institutions were found. Forty-five respondents were evenly selected from the three remaining communities (Telaku Bokazo, Anwia, and Salma), with 15 respondents each from these communities.

The majority of the respondents were males (61.5%) because most of the males are engaged in mining-related activities and other occupations that are important for the study and are deemed to have in-depth knowledge about mining-related issues, as similarly reported by Mabey et al. [8] and Kamga et al. [38]. Most of the respondents fall within the age group of 36–45 constituted 56.9%. Usually, this age group consists of key informants and opinion leaders [8]. Concerning education, most of the respondents (36.9%) had no formal education; this was due to the remoteness of some communities, which deterred most respondents from having access to education at an early age. This was in contrast to the report by Kamga et al. [38] where most of the respondents only attained primary level education, due to the influence of mining. Most of the respondents (50.8%) are engaged in agricultural activities as farming is the dominant activity in the area.

### 3.2. Results of the NDVI Analysis 

In our study, three classes are used in categorizing the values of NDVI images. Class 1 represents areas with low density with values from 0.02 to 0.21, Class 2 has values from 0.21 to 0.26 representing areas with medium density, and Class 3 values are from 0.26 to 0.3+ representing high-density areas. Table 2 shows the NDVI density classes in the years 2008, 2015, and 2020. As presented in Table 2 and Figure 4, significant changes occurred in the medium and the high-density classes, whereas the low-density class saw very slight changes. The category of medium NDVI density (class 2) increased from 34.32% in 2008 to 44.22% in 2015 and then reduced to 33.38% in 2020. Also, the category of high NDVI density (class 3) has decreased from 59.40% in 2008 to about 47.01% in 2015 and increased to 58.89% in 2020. In contrast, the category of low NDVI density has increased from 6.28% in 2008 to 8.76% in 2015 and then reduced to 7.73% in 2020. Figure 5 shows the NDVI density map of the study area in 2008 and 2015, and 2020. Band math was then conducted on the resulting NDVI images by subtracting the 2015 image values from the 2008 image values and 2020 image values from the 2015 image values. The resulting values gave the changes in the vegetation amount and status that occurred between these three time periods for every image pixel. However, in the NDVI change detection, the three classes denote the rate of changes that took place in the area within the 12 years (2002, 2015, and 2020) period. The low class in red indicates negative change, the medium class in yellow indicates no change, and the high class in green shows a positive change.

Due to the difficulties in acquiring the LULC map of the study area (mining host communities), the LULC map of the study region (whole district) was used, but mining host communities and their environs received much attention during the analysis processes. LULC classes were extracted through site visitation and available topographic maps. The maximum likelihood classifier was used to perform a supervised classification, and four land use classes, such as agricultural lands, residential areas, forest reserves, and bare land, were identified. 

Table 3 and Figure 6 present the pattern of land use changes between 2008, 2015, and 2020. Figure 7 represents the LULC images of the study area for 2008, 2015, and 2020. The spatial extent of agricultural lands reduced from 82.13% in 2008 to 80.54% in 2015 and slightly increased to 81.11% in 2020. Forest reserves expanded from 14.23% in 2008 to 15.58% in 2015 and then reduced to 14.10% in 2020. The reduction in the forest reserve was due to the intense increase in mining operational activities between 2013 and 2016. Residential areas, including mining plant sites and community resettlement projects, kept increasing to replace agricultural lands and bare areas in the 12 years mining periods. Residential areas rose from 2.09% in 2008 to 2.16% in 2015 and further increased to 3.85% in 2020. Bare areas also saw a slight increase from 1.55% in 2008 to 1.72 in 2015, and then reduced to 0.94 in 2020, the reduction of the bare areas in 2020 was due to increase in residential lands in the same period. However, comparing the overall percentage changes in LULC between 2008 and 2015 and between 2015 and 2020 periods shows that agricultural lands reduced by −0.98%, which resulted in an increase of 5.21% of the bare area between the 2008 and 2015 periods. Forest reserves and bare area reduced by −4.99% and −29%, respectively, while residential areas/lands increased by 28.17% between the 2015 and 2020 periods.

### 3.3. Field and Community Survey

#### 3.3.1. Impact on Water

Active and shutdown mines are globally noticed to possess severe and lasting threats to water bodies [23,36,37]. Liquid waste released during the processing stage of ores are mostly treated with advanced refinement methods, whiles “secondary” wastewater with fewer concentrations of contaminants like mine drainage water, leaching from tailings dams, and water from waste rock piles are often and migrate directly into nearby water bodies or are inadequately treated in ponds [23,38]. Unlike other large-scale mining companies where mining effluents are directly discharged to nearby rivers and streams [16], Adamus Resources Mines observed good and acceptable waste management practices for solid and liquid waste, including reuse, recycling, well-constructed tailings dams, landfill, and offsite incineration. These waste management practices deter the mining company from directly polluting available water bodies through effluent discharge. However, findings from the field study indicate that significant rivers and streams, such as Subri, Anwiabaka, Bruma, Angajaleh, Kandagale, Bangara, and Kowire, provide ready sources of water for domestic and agricultural purposes to surrounding communities like Nkroful, Anwia, Telaku-Bokazo, Aluku, Salman, and Kandagale are greatly polluted by mine wastewater drainage and heavy run-off from heaped waste rocks at mining pits (Figure 8). These rivers are heavily affected by mine explorations, due to their closeness to mining pits and operational sites.

Some streams in the area are shrinking, due to the constant filling of waste and rocks during run-off from mine sites and alterations by the mining company to divert or change the watercourse to provide enough spaces for mining activities. Interviews with some respondents from Nkroful, Salman, and Teleku-Bokazo revealed that they sometimes found dead fishes at the shores of rivers and streams with toxic effluents from mines sites as causative agents. They often find it challenging to access water free from contaminants for domestic and agricultural activities even though rivers and streams are abundant.

Data from the District Assembly also has it that government, corporate bodies, and individuals invest huge sums of money of about GHȻ 14,000 ($2500) and drill as deep as 90–120 m to have access to uncontaminated underground water in mining host communities. This conciliates with findings by the authors of [39,40] that mining effluents are primarily acidic and saline, with sulfate (SO_4_^2−^), iron (Fe), and metalloids highly concentrated in it, which threatens the lives of habitats and negatively alters the prolificacy of communities.

#### 3.3.2. Impact on Vegetation/Ecosystem and Agricultural Land

According to Andre and Gavin, Ghana’s gold mining industry is more helpful to the national economy. In contrast, indigenes of local or hosting communities face the resulting social and environmental consequences, including the distraction of arable lands [2,41]. The study found out that using sophisticated machine to embark on mining operations, that is, clearing of sites, drilling, and blasting, has resulted in deforestation and a shift in the land use pattern in the southwestern part of the Ellembelle District. Vast portions of vegetative cover have been cleared off as extensive large-scale mining unfolds in affected areas. Mining host communities, such as Salman, Anwia, Telaku-Bokazo, and Nkroful, have experienced vegetation loss with large tracts of land rendered bare, due to gold mining operations. Vegetative cover loss in these areas is associated with repercussions, such as soil erosion, run-off, gullies, less groundwater recharge, and concentration of carbon dioxide (CO_2_) in the atmosphere. Apart from erosion, land devoid of vegetative cover results in loss of viability for agricultural use and loss of habitat for animals [16]. Most respondents expressed their views on how large-scale mining had negatively affected agricultural activities, especially in an interview with a 48-year-old farmer who stated that; “*Before the commencement of Adamus operations, we used to have a lot of animal species in our forest. The most common ones were rabbits, rodents (grasscutters and rats), antelopes, snails, and monkeys. Now, these animals are no more because of the distraction of the forest and heavy operational sounds by the mining company. Again, almost 80% of our vast land for agricultural purposes has been taken by the mining company, and we now find it difficult to have access to land for food production. This has made access to the limited available farmland very competitive and problematic. Too much pressure exerted on limited available farmland and lack of good farming practices makes it difficult for the land to regain its fertility leading to low agriculture output*”. Figure 9 shows a typical landscape before the mining activities and the current destruction with enormous erosion.

A study by the Food and Agriculture Organization confirmed that land degradation and loss of arable land saw a very height in Ghana between 1990 and 2005, due to gold mining resulting in the extinction of 26% of forest cover and 15–20% of arable land at the southern part of the country (Tarkwa, Ayanfuri, Bogoso, and Dunkwa) where gold mining is very active [15,16]. From the field survey, it was observed that several gullies have emerged in most mining sites caused by massive erosions, due to the absence of vegetative cover. Such gullies and some old miming pits are filled with stagnant water serving as breeding places for reptiles like snakes that possess treats to the health of residents [16]. Figure 10 is an old pit containing stagnant water at the Salman-Akango pit concession.

#### 3.3.3. Impact on Land and Soil

Another component of the environment that suffers a devastating effect from gold mining is land/soil. Mining, irrespective of the type and mode of operations, leads to the distraction of land and soil organisms resulting in soil infertility in affected areas. This situation becomes worst if appropriate measures are not effectively enforced [8,16]. Large tracts of land and vegetation cover in most gold mining regions in Ghana have been cleared to provide space for surface mining activities [35]. 

It was noticed during the field study that large-scale gold mining explorations have caused excessive damage to the lands and soil of mining communities within the Ellembelle District. The activities of Adamus Resource Ltd., such as excavation, deep drilling, pitting, and frequent excessive blasting, have caused land degradation and landscape changes (Figure 11A,B) in host communities. Data gathered from the District EPA indicates that about 530 hectares of land have been disturbed by large-scale mining operations and out of which about 110 (20%) hectares have been rehabilitated (Figure 12). Mining host communities, like Nkroful, Salman, Telaku-Bokazo, Anwia, and Aluku, have experienced decreases in the physical soil qualities, due to the constant removal of the fertile topsoil, massive erosion, and sedimentation. Harmful chemicals used during blasting, transportation of toxic substances from mining sites or pits, and the destructions of stable soil aggregate had also caused reductions in the soil quality in these mining host communities [2]. The emergence and concentration of chemicals and microbiological factors in the soil threaten the growth and survival of plants, and therefore, jeopardize agricultural land sustainability in mining areas [27].

It was observed that cash crops like cocoa, coconut, and rubber plantations that were in existence in host communities and their environs before the commencement of gold mining operations are gradually losing lives, with some of such crops currently having yellowish leaves, due to the concentration of harmful chemicals in the soil and the loss of soil fertility to support plants growth. This affirms the work by the authors of [42,43], which postulated that most available plant nutrients are contained by the topsoil, which is estimated to be 20 cm deep, and bulldozers’ clearing impoverish the soil fertility and disclose the subsoil, which has limited fertility for plant survival [44].

#### 3.3.4. Noise Pollution

The noise was highly overlooked by Ghanaians and some authorities from developing countries as a form of pollution until recently where many people started noticing its related problems on humanity [2,5,45]. Today, nearly every industry that employs heavy machinery operates within a given noise pollution standard with regular assessment of noise emission levels by the designated body [2]. Noise pollution is inevitable during mining operations, due to the use of heavy and sophisticated and other inclusive activities like excavation, blasting, drilling, and hauling [1,8]. Field observation and data gathered from the Ellembelle District EPA reveals that host communities experience minimal noise from the Adamus mines processing plant site itself. Communities are usually abused with excessive noise and strong vibrations, mainly during blasting periods at mining sites. The excessive noise and vibrations produced during blasting have affected buildings in host communities, especially in Nkroful and Anwia, with cracks easily seen on structures, due to the regular waggling of buildings. According to the case company and the District EPA, the severity of both noise and vibration produced is always moderate from a scale of low to high, and the ambient noise level is sometimes above standard, mostly at night times. Respondents from Nkroful and Telaku-Bokazo indicated that other primary sources of noise that are of high nuisance are that emanating from heavy trucks when passing through communities to and from mining sites. Besides, noise pollution by mining companies also has intrinsic effects on mineworkers as continuous exposure to loud noise causes hearing problems. A study by the authors of [46] had it that 59 (23%) out of 250 large-scale mining workers in Ghana had noise-induced hearing loss [2,47].

Notwithstanding efforts by authorities and mining companies to curb the effects of noise on local people, the impacts of noise pollution go a long way to affect animals in their natural habitat near mining sites. According to National Geographic Resource (2019), loud sounds above 84 decibels can harm a person’s ear and negatively affect the health and wellbeing of wildlife. A caterpillar is said to have a malfunctioning heart, while the number of chicks of bluebirds reduces when exposed to excessive noise [48]. Some animals like bats, rabbits, and some rodents use typical sounds for several purposes, including navigation, food search, attracting mates, and escaping predators. Noise pollution distracts them from achieving these tasks rendering their inability to survive in such areas [48]. Places and forests closer to the Nzema mines operational sites have lost several species of animals, due to their inability to bear excessive noise emitted during blasting and other mining activities, thereby migrating to favorable environments.

#### 3.3.5. Impacts on Air

The quality of air in the district, especially in active mining communities, has been threatened, due to large-scale surface gold mining activities. The study discovered that air pollution in Nkroful, Telaku-Bokazo, Anwia, Aluku, and Salman emerges from several causative mining activities, with the highly noticeable ones being dust from untarred roads. These roads were used by mining companies for continuous transportation of workers and heavy-duty machines to and from operational sites, smokes/fumes and chemicals from vehicles, and processing plants, dust and chemicals released during excavation, blasting, and loading of ores. The severity of how activities affect air quality is presented in Figure 13 in percentages.

Amponsah (2011) asserted that mining exploration and operation activities possess an inescapable physical and material harm to both the environment and the inhabitants. The environment of mining communities is always at risk, due to its exposition to pollutants like fumes, dust, and chemicals, thereby creating health and safety hazards [18]. It was revealed during the study that some residents of Nkroful demonstrated to the District Assembly during early 2018, due to a minimal outbreak of respiratory ailment in the town, which was linked to severe air pollution by dust from untarred roads used by mining companies. An in-depth interview with a 44-year-old opinion leader lamented that: “*Citizens of Nkroful and Telaku-Bokazo communities are heavily bombarded with dust pollution since the Adamus Mines started operations. This issue of dust pollution became worse when work commenced on the Nkroful and Teleku-Bokazo sites due to their closeness to communities, and that saw to the partial resettlement of project affected persons in both communities. Severe dust emissions occur when heavy-duty trucks and other vehicles ply on untarred communities’ roads. However, the truth must be established that Adamus Resources Ltd. has its mining roads, and therefore its heavy-duty machines seldom ply the community roads. Complaints were lodged by the Project Affected Persons to the Resettlement Committee, which picked them as concerns for discussion with the Company. These complaints were rampant at the initial stage before the resettlement of the affected Persons. Such discussions, coupled with several demonstrations to the District Assembly, have led to dust reduction exercises carried out periodically by the company on the community’s major road. The problem still keeps persisting*”.

## 4. Conclusions and Recommendation

Findings from both the NDVI analysis and the field survey show that large-scale gold mining operations threaten the environment of the study area. Within the twelve years (2008 to 2020) considered for our analysis, empirical findings show that severe environmental impacts and changes took place between 2013 to 2016 when mining activities were very intense in the area. However, aside from the already discussed negative effects emanating from large-scale mining activities in the area, some issues like weak enforcement of mining policies, ineffective collaborations between mining support and environmental protection institutions, and limited community participation in environmental decision-making processes were noticed during the study. Although the case company is continuously modifying its operations processes to control its resulting impacts on host communities and the environment, such interventions are inadequate to curb the resulting environmental impacts. Mining host communities and their environment severely bear the consequences that arise from mining. Thus, the following recommendations are made to improve sustainable mining, enhance environmental sustainability, and achieve sustainable development in the district and in Ghana:*Ensuring Effective Stakeholder Coordination*, almost all environment components suffer the consequences of large-scale mining activities; therefore, controlling such impacts requires cross-sectoral efforts from all mining sector institutions. There should be effective coordination among all environmental protection and mining support institutions like EPA, Mineral commission, Forestry, and other stakeholders, such as traditional councils and opinion leaders of host communities. Doing this will offer traditional rulers a golden opportunity to add their views to decisions that directly or indirectly affect their wellbeing. Effective collaboration with well-defined roles among these stakeholders will provide a solid body to ensure environmental sustainability by effectively monitoring and scrutinizing the objectives and activities of mining companies. *Frequent Environmental Education in Mining Communities*, environmental protection institutions like the EPA should frequently organize environmental education and awareness programs for mining companies and citizens of host communities on the value of the environment and how to enhance its sustainability as mining operations unfold. Mining companies should be made aware and be educated on some of their unwanted practices and the associated negative impacts that jeopardize environmental sustainability.*Conducting of Thorough Environmental Impact Assessment,* as the mining company is still searching for gold deposits to extend its operations to other areas in the district, appropriate and responsible agencies like the EPA are to conduct an extensive environmental impact assessment to determine and weigh the potential hazards of a newly proposed site before granting a permit. The mining company should be denied a permit if these potential hazards to the environment are very severe after the impact assessment.*Embarking on Afforestation Programs,* the government should make it mandatory for large-scale mining companies to incorporate afforestation programs in their activities. Abandoned mine sites and degraded land should not only be rehabilitated by filling pits with gravel, but should include replanting trees to regain the lost vegetation.*Installations of Noise Absorbing Machines and Formulation of Strict Blasting Regulations*, mining companies should install modern noise-absorbing machines in host communities to help absorb and reduce the excessive noise generated into the community during blasting and other operational activities. Blasting regulations should also involve a defined distance from mining operating sites to communities to deter blasting at mines sites near communities.

The research was conducted on a single large-scale mining company operating in four communities within one district in Ghana, so it is advisable to be careful about the generalization of our results to activities of other large-scale mining companies in different regions throughout the country. However, the literature shows that the reported results are reconcilable with other large-scale mining activities in the country and other developing countries. The LULC maps and the NDVI data used for the analysis were based on the whole study region (District) instead of the study areas (host communities), due to the difficulties in acquiring LULC images for the study areas. For that matter, it would be interesting for further studies to be conducted by using GIS and remote sensing methodologies. This would quantify and assess the environmental impacts emanating from both small-scale and large-scale mining in most dominants mining regions of the country, and that will help to detect the exact or direct changes and impacts that are solely caused by mining.

## Figures and Tables

**Figure 1 ijerph-18-07044-f001:**
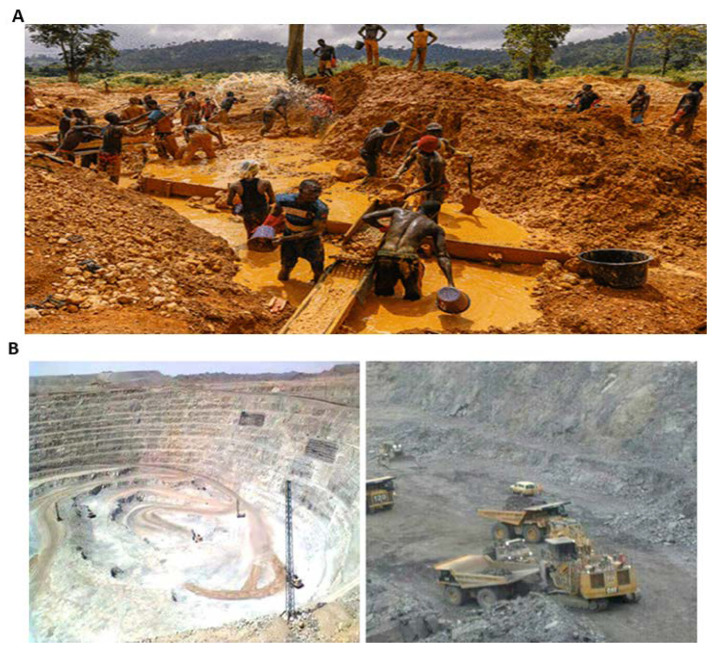
(**A**) Small-scale (galamsey) mining activities, (**B**) large-scale surface mining activity, August (2020).

**Figure 2 ijerph-18-07044-f002:**
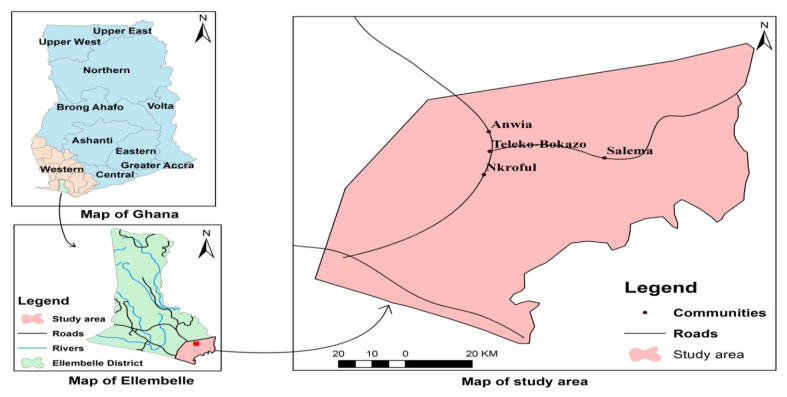
Map of Ellembelle District containing study communities (2020).

**Figure 3 ijerph-18-07044-f003:**
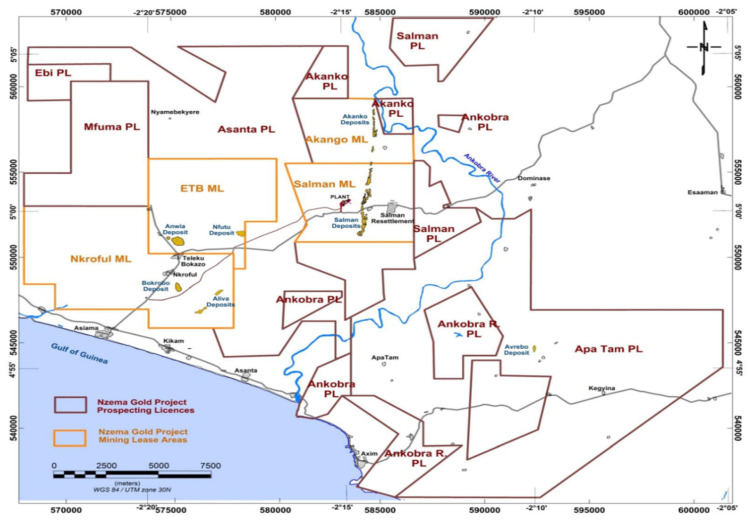
Mineral license perimeters, site layout, and deposit locations (2013).

**Figure 4 ijerph-18-07044-f004:**
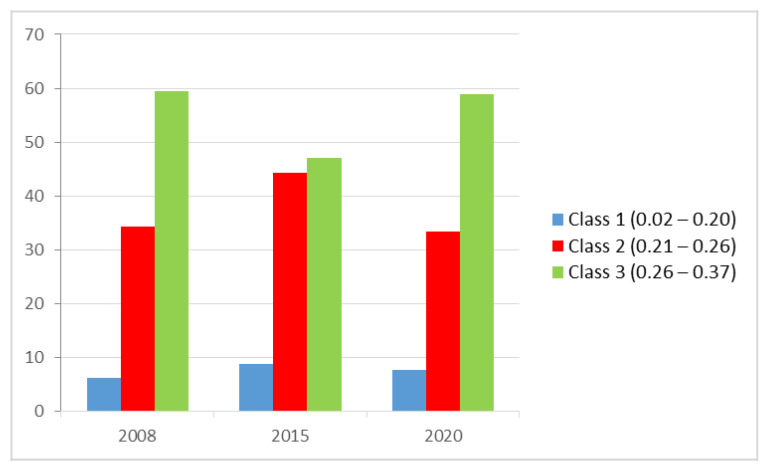
NDVI density classes changes during the period of 2008 to 2020.

**Figure 5 ijerph-18-07044-f005:**
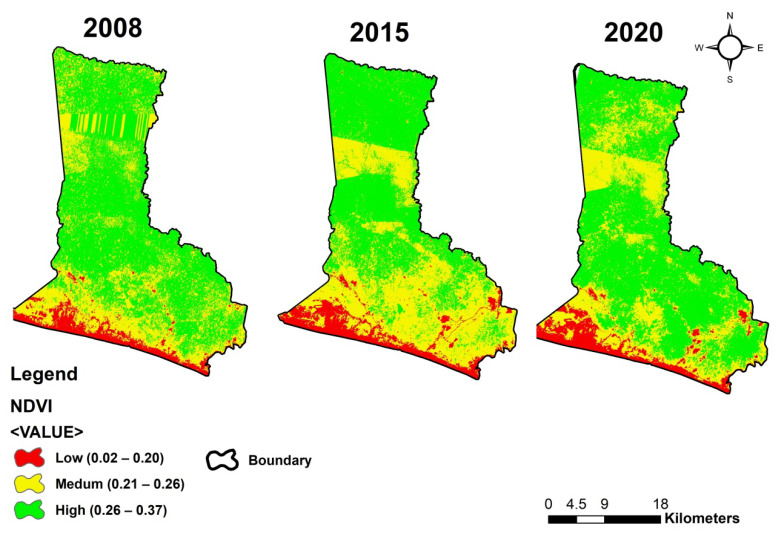
NDVI density maps for 2008, 2015, and 2020.

**Figure 6 ijerph-18-07044-f006:**
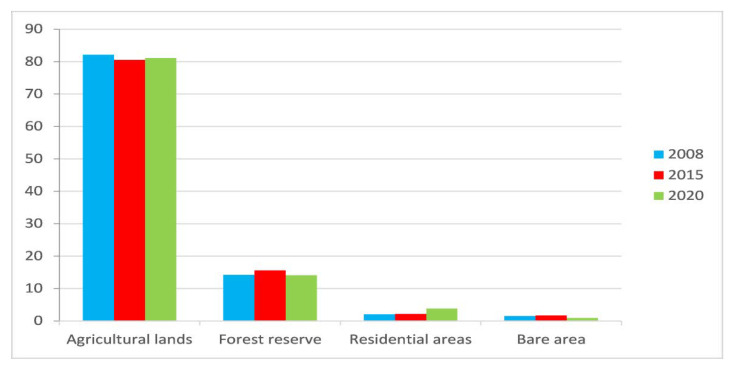
The land covers categories changes during the 2008 to 2020 periods (%).

**Figure 7 ijerph-18-07044-f007:**
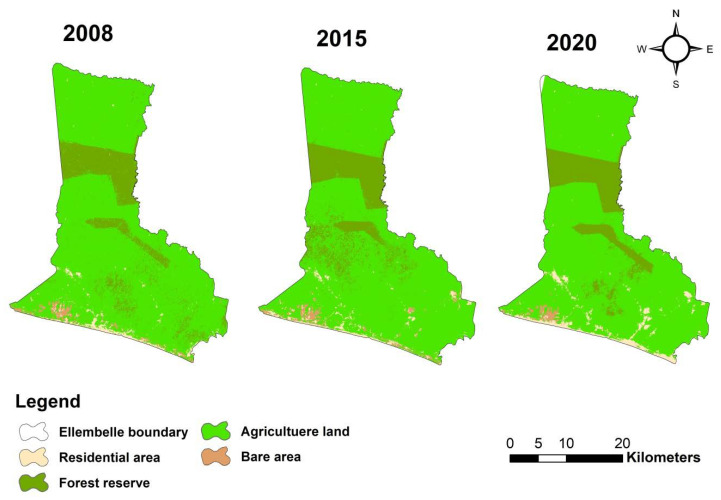
LULC classified images of the study area for 2008, 2015, and 2020.

**Figure 8 ijerph-18-07044-f008:**
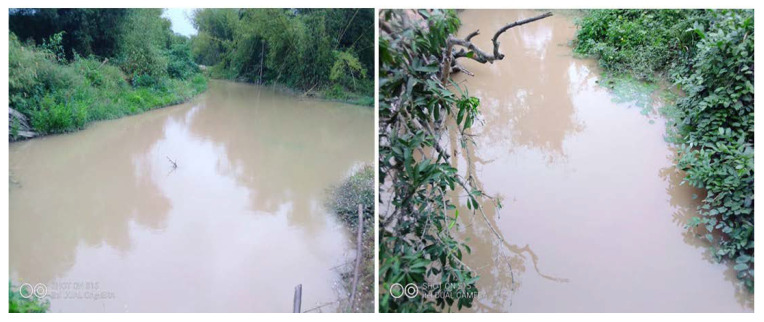
Subri River and Broma River are heavily polluted by large-scale mining activities (effluent) in Ellembelle, August (2020).

**Figure 9 ijerph-18-07044-f009:**
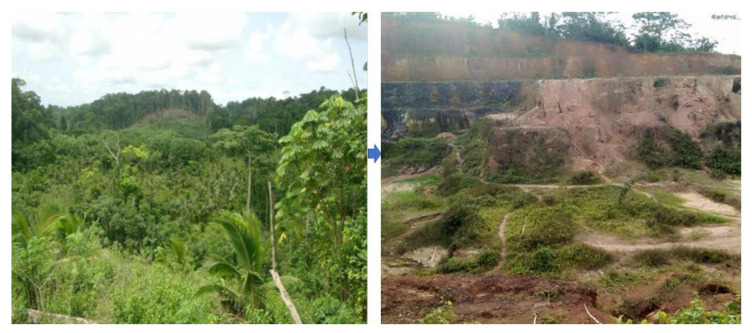
Destructed vegetation and land with massive erosion taking place, August (2020).

**Figure 10 ijerph-18-07044-f010:**
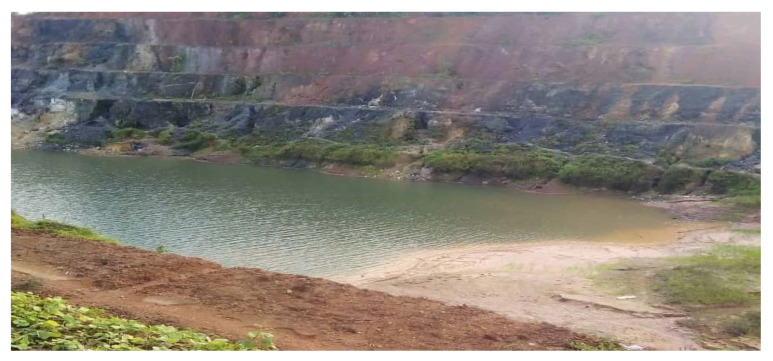
An old pit filled with stagnant water (Salman-Akango pit), August (2020).

**Figure 11 ijerph-18-07044-f011:**
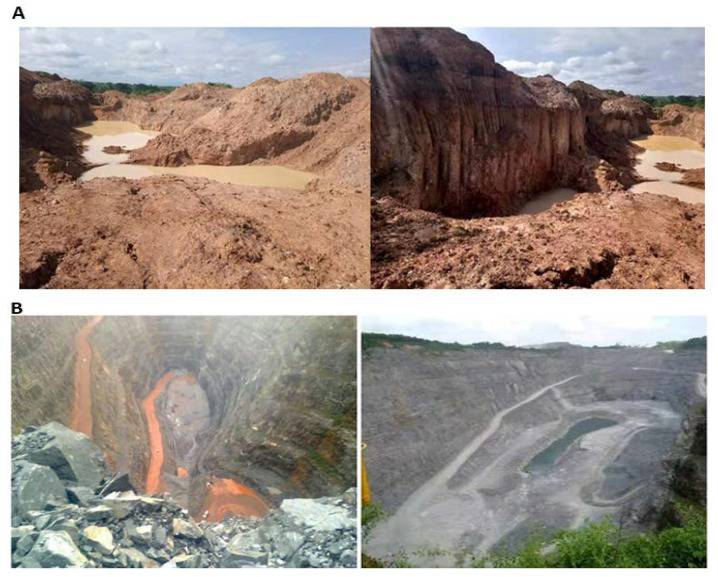
(**A**) Degraded land due to mining activities (excavation), (**B**) blasting and pitting (Adamus pit) caused massive land degradation, August (2020).

**Figure 12 ijerph-18-07044-f012:**
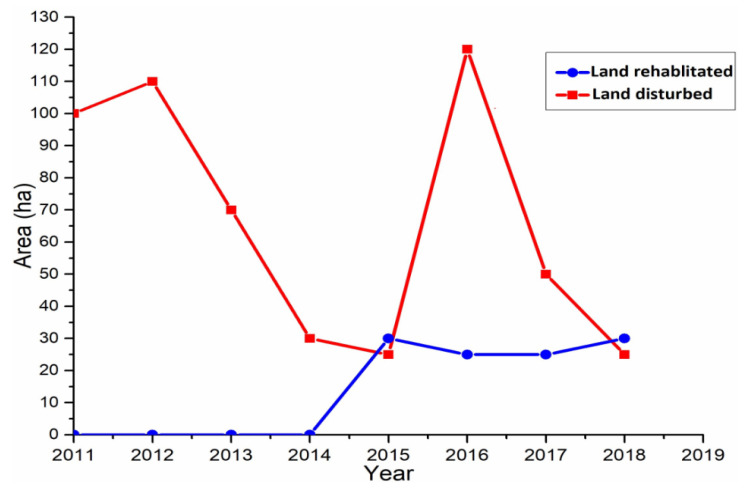
Land disturbance and rehabilitation in the district (based on data from the District EPA).

**Figure 13 ijerph-18-07044-f013:**
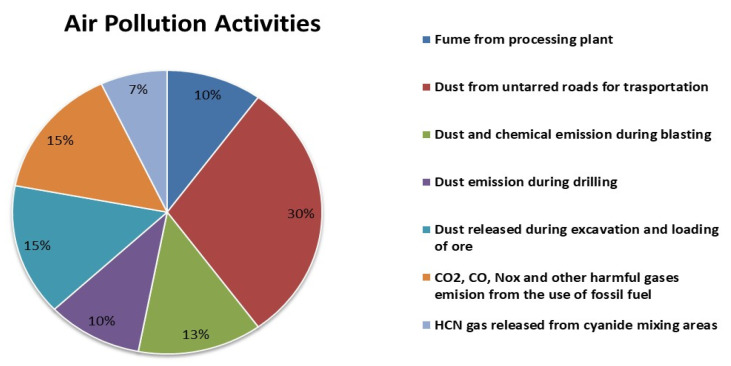
Percentage presentation of air pollution activities (based on data from the District EPA).

**Table 1 ijerph-18-07044-t001:** Demographic characteristics of respondents.

Variable		Community			Total
	Nkroful	T. Bokazo	Anwia	Salma	*n* = 65 (%)
**Gender**					
Male	13	7	10	10	40 (61.5)
Female	7	8	5	5	25 (38.5)
**Age group**					
20–35	4	3	3	4	14 (21.5)
36–45	11	8	10	8	37 (56.9)
46+	5	4	2	3	14 (21.5)
**Educational level**					
Non-Formal	3	5	8	8	24 (36.9)
Basic	4	3	3	1	11 (16.9)
Secondary	5	4	4	3	16 (24.6)
Tertiary	8	3	0	3	14 (21.5)
**Occupation**					
Stakeholder institution	3	0	0	0	3 (4.6)
Opinion leader	3	2	2	2	9 (13.8)
Mine worker	2	1	0	1	4 (6.2)
Farmer	8	8	9	8	33 (50.8)
Others	4	4	4	4	16 (24.6)

**Table 2 ijerph-18-07044-t002:** Change of the NDVI density classes between 2008, 2015, and 2020.

NDVI (KM^2^)	2008		2015		2020	
	km^2^	%	km^2^	%	km^2^	%
Class 1 (Low)	59.59	6.28	83.22	8.76	73.40	7.73
Class 2 (Medium)	325.76	34.32	419.68	44.22	316.78	33.38
Class 3 (High)	563.80	59.40	446.25	47.01	558.97	58.89
Total	949.15	100	949.15	100	949.15	100

**Table 3 ijerph-18-07044-t003:** Comparison of areas and pattern of changes in LULC classes between 2008, 2015, and 2020.

LULC Type	2008 LULC Area		2015 LULC Area		2020 LULC Area		Change between 2008 and 2015		Change between 2015 and 2020	
	km^2^	%	km^2^	%	km^2^	%	km^2^	%	km^2^	%
Agricultural lands	779.55	82.13	764.44	80.54	769.91	81.11	−15.11	−0.98	5.47	0.36
Forest reserve	135.10	14.23	147.89	15.58	133.84	14.10	12.79	4.52	−14.05	−4.99
Residential areas	19.80	2.09	20.50	2.16	36.58	3.85	0.70	1.74	16.08	28.17
Bare area	14.73	1.55	16.35	1.72	8.85	0.94	1.62	5.21	−7.50	−29.76
Total	949.18	100	949.18	100	949.18	100	-	-	-	-

## Data Availability

The data presented in this study are available on request from the corresponding author.

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
