# Peer review of "An Integrated Approach to Assess the Environmental Impacts of Large-Scale Gold Mining: The Nzema-Gold Mines in the Ellembelle District of Ghana as a Case Study"

_ijerph, 2021, doi:10.3390/ijerph18137044_

Round 1
Reviewer 1 Report
This manuscript offers an analysis of large-scale gold mining in a portion of the nation of Ghana. The authors begin with an Introduction that is divided into two parts. In the first part, they describe the economic importance of gold mining to developing nations in general and to Ghana in particular, while pointing out that mining activity has led to a great deal of environmental damage in the study area. In the second part they give a more detailed description of mining activities and include photographs of mining sites. I think the Introduction is well done in its narrative and focus.
In Part II, Materials and Methods, the authors begin with a full description of the study area, the Ellembelle District. They provide a set of maps to help the reader place the location within Ghana. The authors follow that with a section on the enterprise conducting mining activities and then proceed to data collection, introducing the The Normalized Difference Vegetation Index.
Section III features results and discussion. The authors show land use changes for three time periods, 2008, 2015 and 2020 for four categories: agricultural land, forest, bare areas and residential. These results showed a reversal in trends in each land use type for the two intervals. The authors use a set of maps and tables to present these results.
Next, the authors discuss findings from the field and community survey. They initially give the results on effects on water, concluding that even with the presence of infrastructure to prevent effluent from polluting rivers, the pollution level in the local rivers is extremely high, and they attribute this to both the large scale and smaller scale mining operations. They present photographs of two local rivers to demonstrate.
From this point the authors discuss pollution effects on lands such as agricultural and forest lands. This discussion is extensive and again contains numerous figures. The authors complete the results by discussing air pollution and noise pollution associated with mining.
Section IV is Conclusion and Recommendation. This section is very well organized, staring off with a general summary and then moving on to five specific items that are concise but thorough at the same time.
The manuscript has a total of 39 references.
I found this manuscript very interesting, and I think it will make a useful contribution to the journal. The authors appear to be thorough and transparent throughout the manuscript and the topic of the environmental effects of mining is one of the most significant and relevant topics of our era. I believe the manuscript should be published.
There are however, many problems in the manuscript with poor English grammar. The manuscript needs a thorough review by someone who is competent in English to correct these flaws before the manuscript can be published.
Examples: line 273, whiles should be while
Line 274 mines should be mine (yes the drainage water comes from more than one mine, but the singular “mine should be used here because it describes a type of drainage water)
Line 288 mines exploration should be mine (same reason as above) - the authors got this right on line 292 with the singular mine (describing sites)
Line 295 fishes should be fish, that’s just the way the English language works – the plural for fish is also fish, not fishes
The sentence starting on line 49 is problematic “It immense ……” Maybe that should be Its
Line 62 increasingly should be increasing
Line 64, “stood” should be “have received”
Line 70 utilized should be utilize
Delete the word “the” in line 71
Line 83 ensures should be ensure
The list of these minor errors could go on and one. They are annoying for the reader and if the manuscript were published with them in it, this would not reflect well on the journal. I also want to point out that there were a few places in the manuscript where there were subject verb agreement problems (a singular subject with a plural verb or vice versa).
Once these numerous grammar glitches are cleared up, the manuscript should be published in my opinion. On substance, it is quite good.
Author Response
We thank you for the opportunity granted us to revise our manuscript. Your constructive comments and suggestions have substantially improved our manuscript. Based on these comments and suggestions, we have carefully modified the original manuscript. The “Track Changes” function with red color was used to respond to the reviewers’ comments in the revised manuscript for easy identification. The manuscript has been thoroughly edited by a professional English editor and comments are addressed in the same order as mentioned in the review.
Please attached is the point-to-point response to your comments.
Thanks

Reviewer 2 Report
The article concerns an important issue – environmental impacts of large-scale gold mining as a case study.
The information contained in the article is valuable, because they have practical application. The idea for research is good. However there are some issues, which needs to be improved in the article, hence I recommend the article to publish after a major revision.
1. The most important research results (numerical values) should be included in the abstract.
2. It is necessary to describe the survey researchs in more detail (additional chapter) - characteristics of the research group, statistics, questions, survey and questionnaire patterns should be attached to the article as a supplement.
3. This article does not include a state-of-the-art review made in a sufficient way. The literature review is very poor. The authors does not indicate what the research and results included in the article are novel for science described in the literature.
4. The analysis of the obtained results is insufficient, there are no deeper reflections of the research resulting
5. The conclusions and recommendations do not result from the conducted research (e.g. lines 436-438; recommendations 1 and 2).
6. Figure 4 duplicates the content of table 1 and figure 6 duplicates the content of table 2. Please leave the table or figure with described.
7. "Large-scale gold mining" and "small-scale gold mining" should be clearly defined.
8. There are too few references to the survey results.
9. Please provide quantitative characteristics of the pollution of reservoirs and watercourses.
Overall, I rate this article as average but due to the importent subject, it has potential. The article needs improvements and additional explanations/desciptions. The current version of the article cannot be printed. It is necessary to improve it according to above-mentioned remarks.
Author Response

(The authors gave the same response as above.)

Reviewer 3 Report
I found the topic of the study very interesting and in line with the scope of the journal. To improve the overall quality of the manuscript, I have some suggestion/comments as below:
The quality of the figures 2, 3 may be improved, at least in my pdf they are getting a bit distorted.
Lines 190-195: The Normalized Difference Vegetation Index (NDVI), need a better explanation on the equation 1. It is hard to understand it.
Lines 196-211: It is recommended to insert a diagram or graph to explain the different stages to map gold mining-related vegetation cover changes in Ellembelle, from images for 2015 and 2020 from archives of the United States Geological Survey 198 (USGS). It is hard to understand it.
Need a better explanation in table 2. It is hard to understand it and you should comment on the Comparison of areas and pattern of changes in LULC classes between 2008, 2015 and 2020.
References: bibliographic citations should be reviewed (format of the year, ...).
English needs to be revised.
Author Response

(The authors gave the same response as above.)

Round 2
Reviewer 2 Report
The explanations provided by the authors are insufficient. Authors should improving the article regarding following comments:
Comment 2. It is necessary to describe the survey research in more detail (additional chapter) - characteristics of the research group, statistics, questions, survey and questionnaire patterns should be attached to the article as a supplement.
Comment 3. The analysis of the obtained results is insufficient, there are no deeper reflections of the research resulting.
Comment 7: There are too few references to the survey results.
Author Response
We thank you for the opportunity granted us to revise our manuscript. Your constructive comments and suggestions have substantially improved our manuscript. Based on these comments and suggestions, we have carefully modified the original manuscript. The “Track Changes” function with red color was used to respond to the reviewers’ comments in the revised manuscript for easy identification. find attached a letter of our response.

Reviewer 3 Report
The revised version is well-written, scientifically conducted and the conclusions were comprehensively supported by the data, therefore, the revised version can be accept in present form.
Author Response
We thank you for the opportunity granted us to revise our manuscript. Your constructive comments and suggestions have substantially improved our manuscript. Based on these comments and suggestions, we have carefully modified the original manuscript. The results have been modified to give a clear presentation and the English language is edited, well-styled, and the manuscript is well arranged following the journal's rules. The “Track Changes” function with red color was used to respond to the reviewers’ comments in the revised manuscript for easy identification.
We appreciate your time and good work to shape our manuscript.